# Comparison of SP, SMAT, SMRT, LSP, and UNSM Based on Treatment Effects on the Fatigue Properties of Metals in the HCF and VHCF Regimes

**Rui Chen, Hongqian Xue * and Bin Li**

School of Mechanical Engineering, Northwestern Polytechnical University, Xi'an 710072, China;
2021180032@mail.nwpu.edu.cn (R.C.); bin.li@mail.nwpu.edu.cn (B.L.)
* Correspondence: xuedang@nwpu.edu.cn

**Abstract:** This paper aims to provide a better understanding regarding the effects of shot peening (SP), surface mechanical attrition treatment (SMAT), laser shock peening (LSP), surface mechanical rolling treatment (SMRT), and ultrasonic nanocrystal surface modification (UNSM) on the fatigue properties of metals in high-cycle fatigue (HCF) and very-high-cycle fatigue (VHCF) regimes. The work in this paper finds that SMRT and UNSM generally improve the high-cycle and very-high-cycle fatigue properties of metals, while SP, SMAT, and LSP can have mixed effects. The differences are discussed and analyzed with respect to the aspects of surface finish, microstructure and microhardness, and residual stress. SMRT and UNSM generally produce a smooth surface finish, while SP and SMAT tend to worsen the surface finish on metals, which is harmful to their fatigue properties. In addition to inducing a plastic deformation zone and increasing microhardness, surface treatments can also generate a nanograin layer and gradient microstructure to enhance the fatigue properties of metals. The distribution of treatment-induced residual stress and residual stress relaxation can cause mixed effects on the fatigue properties of metals. Furthermore, increasing residual stress through SP and SMAT can cause further deterioration of the surface finish, which is detrimental to the fatigue properties of metals.

**Keywords:** shot peening; SMAT; LSP; SMRT; UNSM; very-high-cycle fatigue

## 1. Introduction

In recent years, methods for improving the high-cycle and very-high-cycle fatigue properties of metals have remained an area of focus due to high interest from industries. Cold working surface treatment is one of the most common ways to improve the fatigue properties of metals. Many researchers have studied the effects of different surface treatments on the fatigue properties of various metals in the HCF and VHCF regimes. These surface treatment methods include shot peening (SP), surface mechanical attrition treatment (SMAT), ultrasonic shot peening (USSP), laser shock peening (LSP), surface mechanical rolling treatment (SMRT), deep rolling (DR), ultrasonic impact treatment (UIT), and ultrasonic nanocrystal surface modification (UNSM). Since most research has only focused on the effects of one or two surface treatment methods on the fatigue properties of selected metals, it would be valuable to have a comparison of the noted treatment methods to better understand the similarities and differences as well as the limitations and advantages of the processes.

The fatigue properties of metals are improved by delaying fatigue fractures. In the HCF and VHCF regimes, the fatigue fracture of metal can originate from either the surface crack or interior crack locations. The HCF regime typically refers to the region between $10^4$ and $10^7$ fatigue load cycles, whereas the VHCF regime typically refers to the region beyond $10^7$ fatigue load cycles. Metal materials generally have higher fatigue stress limits in the HCF regime than in the VHCF regime. At higher fatigue stress levels, the surface crack

will initiate and grow if the surface stress intensity factor (SIF) range, $\Delta K_s$, is higher than the surface threshold intensity factor $\Delta K_{th,s}$. Rough surface and small surface defects can increase the stress concentration, which promotes surface crack initiation and growth [1–3]. As the fatigue stress level is lowered, the surface SIF range, $\Delta K_s$, will be less than $\Delta K_{th,s}$ at a certain point, and surface cracks will not initiate. Instead, the failure mode will shift to interior crack failure from that fatigue stress level and lower [2]. With interior crack failure due to internal defects, a fine granular area (FGA) is often observed from the fracture surface. The FGA is a stress concentration area. It is formed as the metal undergoes a very long period of fatigue cycles at low stress levels; thus, it tends to occur in the VHCF regime. After the formation of the FGA, the interior crack starts to initiate and grow, and a fish-eye area can form on some metals [2,4]. Similar to the FGA, the white rough area (WRA) is an internal stress concentration area that can be observed if the fracture is initiated from the locations of easily breakable particles within the material [5,6]. Another type of interior crack failure is the transgranular quasi-cleavage fracture mode. In this type of failure, research from Wang et al. [7] showed that fracture originating cracks are initiated from subsurface facets. As shown above, the overall high-cycle and very-high-cycle fatigue properties of metals can be affected by the surface crack failure and interior crack fracture processes. Surface crack and interior crack failures can lead to significant differences in fatigue lives. Surface treatments strengthen or weaken the fatigue properties of metals in the HCF and VHCF regimes by promoting or delaying surface crack and interior crack fractures.

This paper compares and reviews the effects of selected surface treatment methods on the high-cycle and very-high-cycle fatigue properties of various steel, aluminum, and titanium metals. First, the detailed processes of SP, SMAT, LSP, SMRT, and UNSM are introduced. Next, the paper reviews the effects on the high-cycle and very-high-cycle fatigue properties of metals induced by the selected surface treatment methods, which are discussed in groups based on the results from previous research. In the following sections, the paper discusses and reviews the other changes induced by the selected surface treatments, including surface finish, surface microstructure and microhardness, and compressive residual stress. From these treatment-induced changes, we attempt to identify the causes for the different treatment effects on the fatigue properties of metals in the HCF and VHCF regimes. Finally, some conclusions are made to highlight key observations and findings.

## 2. Treatment Processes

### 2.1. SP

SP is a common surface treatment method used by many industries to improve the mechanical properties of materials. During the treatment process, the workpiece surface is struck by small shots (usually made of metal, ceramic, or glass) at high velocity; the impact forces of the shots cause the plastic deformation of the surface material. Induced by the plastic deformation, compressive residual stresses increase in the surface region, and tensile residual stresses increase in the interior of the material [8,9]. Additionally, the shot peening treatment also affects the surface finish and changes the mechanical properties of the material in the surface region. The common configuration variables of the shot peening process include the shot material, shot size, shot peening velocity, shot peening angle, nozzle distance, and shot peening coverages [10,11]. The configuration can be optimized based on the properties of the workpiece for the best shot peening results. Although shot peening has already been widely applied in industry, specific shot peening approaches, such as severe shot peening and microshot peening, are still being studied to identify ways to improve treatment results. Severe shot peening uses severe parameters for the treatment process, and microshot peening uses micrometer-sized shots to perform the treatment [12,13].

### 2.2. LSP

LSP, also known as laser peening, is a surface treatment process that uses a pulsed laser beam to modify the surface properties of a workpiece (reference Figure 1). A common procedure for LSP treatment is as follows: The workpiece surface is first coated with a sacrificial layer to protect the target surface during the treatment. A layer of running water covering the target surface is then added as a transparent constraint layer. When the pulsed laser beams are fired onto the target surface, the laser beams pass through the transparent constraint layer and strike the sacrificial layer. The pulsed laser then changes the sacrificial layer material into expanding plasma, which is trapped between the constraint layer and the workpiece. The expansion of the plasma then produces a shock wave that moves into the workpiece, causing material plastic deformation. This shock wave changes the material's structure in the surface region and affects the properties of the adjacent material [14,15]. Meanwhile, the material surface roughness also tends to increase as a result of the LSP treatment. The LSP process can be optimized for the pulse energy, overlay rate, pulse duration, constraint layer, and sacrificial layer based on the material of the workpiece.

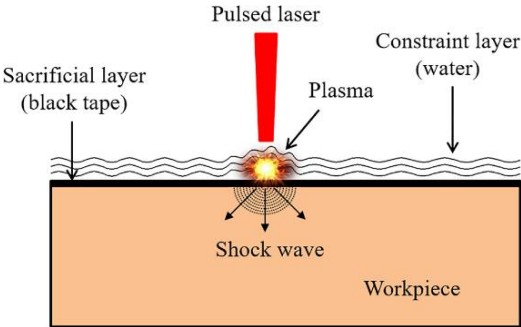

**Figure 1.** Illustration of LSP treatment. Reprinted with permission from Ref. [14]. Copyright 2021 Elsevier.

### 2.3. SMAT

SMAT is a relatively new cold working process derived from the conventional shot peening treatment. In a basic SMAT setup (reference Figure 2), the workpiece being treated is secured in a closed chamber containing shots. The shots for the SMAT process are usually a few millimeters in diameter, which is much larger than the common shots used in shot peening. A vibration generator located at the bottom of the chamber propels the shots to impact the workpiece randomly from different angles. The vibration generator can operate between 50 Hz and 20 kHz [16]; the treatment process is also known as ultrasonic shot peening (USSP) when the vibration frequency is above 16 kHz. Furthermore, the generator power and treatment duration are also adjusted as required for the desired results. Similar to shot peening, the surface material of the workpiece becomes plastically deformed due to the repetitive shot impacts. The plastic deformation leads to changes in the microstructure and mechanical properties of the surface region of the material. A gradient structure with a nanosized grain layer on the material surface can be formed. Additionally, the surface roughness of the workpieces will increase after the SMAT treatment, and surface defects can be induced if the treatment is severe [5].

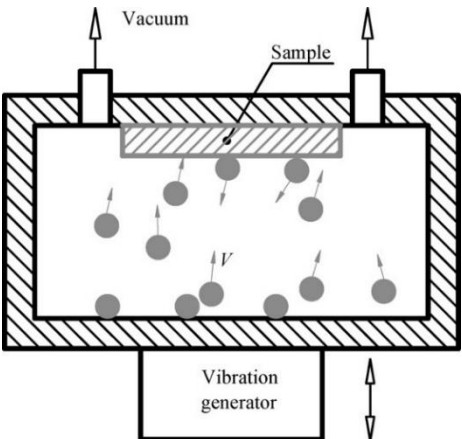

**Figure 2.** Illustration of SMAT treatment. Reprinted with permission from Ref. [17]. Copyright 2008 Elsevier.

*2.4. SMRT*

SMRT and DR are the same surface rolling treatment method. Common materials that can be processed with surface rolling treatments are titanium alloy, carbon steel, stainless steel, and aluminum alloy. The tests are typically performed using cylindrical or flat hourglass-shaped workpieces. There are different ways to perform a surface rolling treatment depending on the material type and geometry. For treating a cylindrical hourglass-shaped workpiece, one common surface rolling method involves pressing and rolling a single polished tungsten carbide (WC/Co) ball over the targeted surfaces while the workpiece is rotating (reference Figure 3). Delgado et al. [18] also described a similar surface rolling setup using a surface rolling machine, where three balls spaced 120° apart—instead of one ball—were used to treat the hourglass specimen surfaces. Saalfeld et al. [19] used another type of surface rolling tool to treat test specimens and added an induction heater to heat the specimens while performing the rolling process. Oevermann et al. [20] used the same surface rolling tool as Saalfeld et al. [19] but applied liquid nitrogen to maintain cryogenic conditions while treating test specimens. For non-symmetrical workpieces, the material can still be treated with surface rolling by using a moveable ball or roller to press against the treated surface [21].

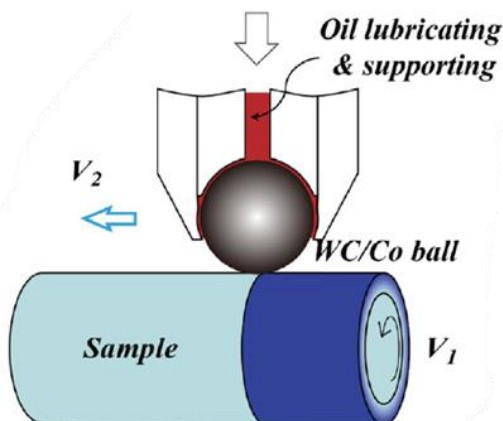

**Figure 3.** Illustration of SMRT treatment. Reprinted with permission from Ref. [22]. Copyright 2015 Elsevier.

*2.5. UNSM*

UNSM treatment (also known as UIT) is an advanced surface treatment method that is effective in improving material fatigue properties. During the UNSM process, a semi-spherical tooltip (usually made from tungsten carbide) is pressed against the target surface

by a static load. Additionally, a vibration generator drives the tooltip up and down at an ultrasonic frequency, such that the tooltip continuously strikes the material surface as it scans through the targeted area (reference Figure 4) [23,24]. Therefore, this method is also commonly referred to as ultrasonic peening treatment or ultrasonic impact treatment. The combined static load and peening impact force from the tooltip causes significant plastic deformation to the material surface and induces compressive residual stress. The plastic deformation also produces a gradient microstructure with a nanograin surface layer and changes the mechanical properties of the material in the deformed region. Moreover, the UNSM treatment can also improve the material surface roughness and reduce surface defects [23]. Common control parameters for the process include ultrasonic amplitude, static load, scanning spacing, and scanning speed.

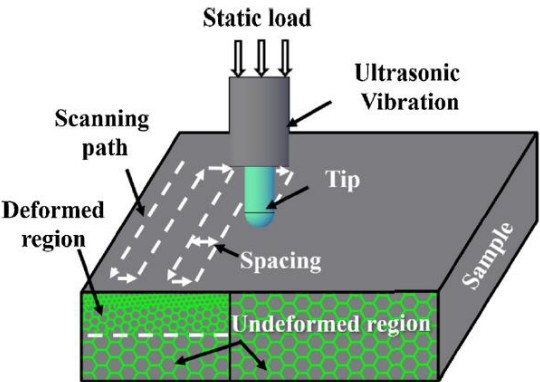

**Figure 4.** Illustration of UNSM treatment. Reprinted with permission from Ref. [23]. Copyright 2020 Elsevier.

## 3. Effects on HCF and VHCF Properties

### 3.1. Effects of SMRT and UNSM

Based on previous research, SMRT and UNSM methods generally improve the fatigue properties of metals in HCF and VHCF regimes. Furthermore, these two methods also tend to cause the shift from surface crack failure to internal crack failure to occur at higher fatigue stress levels.

The rotary bending fatigue test results reported by Sakai et al. [4] showed that surface rolling treatments improved the high-cycle and very-high-cycle fatigue properties of JIS SUJ2 steel specimens, and the improvement was greater in the HCF regime. Additionally, internal crack failures occurred at significantly higher fatigue stress levels on rolling-treated specimens compared to untreated specimens at a load cycle $N_f = 10^7$. In the study by Saalfeld et al. [19], the fatigue test results showed that SMRT significantly improved the fatigue properties of SAE 1045 steel specimens in the HCF and VHCF regimes in a uniaxial fatigue test. The research from Dong et al. [1] indicated that SMRT had strong positive effects on axial-load fatigue properties of friction-stir welded (FSW) 7075-T651 aluminum specimens in HCF and VHCF regimes. The failure mode in the VHCF regime also shifted from surface crack failures to interior crack failures after SMRT (reference Figure 5).

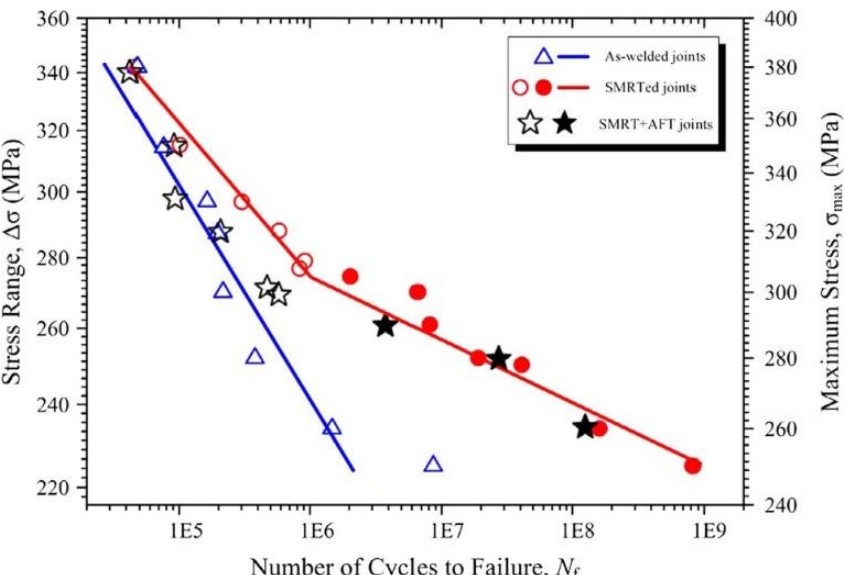

**Figure 5.** Comparison of fatigue test data (axial load, *R* = 0.1) of FSW 7075-T651 aluminum specimens with as-welded, SMRTed, and SMRT + AFT conditions. The SMRT + AFT condition indicates that specimens had an additional microgroove added after processing with SMRT to simulate the surface condition of as-welded specimens. The open symbols represent the surface crack failures and the solid symbols represent the interior crack failures. Reprinted with permission from Ref. [1]. Copyright 2019 Elsevier.

According to studies by Khan et al. [25], He et al. [26], and Cao et al. [27], uniaxial ultrasonic test results have shown that UNSM (or UIT) has significant positive effects on the high-cycle and very-high-cycle fatigue properties of AISI 310 stainless steel, FSW 7075-T651 aluminum, and Ti-6Al-4V titanium specimens, respectively. In addition, Khan et al. [25] and Cao et al. [27] observed a surface crack failure mode in the VHCF regime before UNSM treatment, which shifted to an interior crack failure mode after UNSM treatment. Suh et al. [24] performed rotary bending fatigue tests on UNSM-treated JIS SCM435 steel specimens in the HCF and VHCF regimes, and the test results showed that the fatigue strengths of the specimens continued to rise with increases in the static load of the UNSM treatment (Table 1). The research by Karimbaev et al. [28] showed that UNSM treatments improved the high-cycle and very-high-cycle fatigue properties of AISI 4340 steel specimens more than SP or SP + UNSM treatments in a rotary bending fatigue test (Table 2).

**Table 1.** Comparison of the improvement of fatigue endurance limit (from rotary bending test) and surface residual stress after processing JIS SCM435 steel specimens with UNSM at different treatment static loads [24].

| UNSM Static Load Level | Improvement of Fatigue Endurance Limit at $10^9$ Cycles | Surface Residual Stress (MPa) |
|---|---|---|
| 40 N | 10% | −611.6 |
| 70 N | 20% | −649.4 |
| 100 N | 30% | −685.9 |

**Table 2.** Comparison of specimen conditions, surface roughness, average specimen fatigue lives at 400 MPa, and fatigue limits at $10^8$ cycles for AISI 4340 steel specimens in as-received condition and after SP, UNSM, and SP + UNSM treatments. Measurements for fatigue lives and fatigue limits were based on rotary bending fatigue test results [28].

| Specimen Condition | Surface Roughness | | Fatigue Life at 400 MPa (Cycles) | Fatigue Limit at $10^8$ Cycles (MPa) |
| --- | --- | --- | --- | --- |
| | *Ra* (μm) | *Rz* (μm) | | |
| As received | 1.1 | 3.4 | $4.65 \times 10^5$ | 275 |
| SP | 1.2 | 3.8 | $9.1 \times 10^6$ | 325 |
| SP + UNSM | 0.4 | 1.5 | $16.5 \times 10^6$ | 350 |
| UNSM | 0.3 | 1.6 | $10 \times 10^7$ | 400 |

*3.2. Effects of SP, SMAT, and LSP*

Unlike the SMRT and UNSM methods, the SP, SMAT, and LSP surface treatment methods have shown mixed effects on the fatigue properties of metals. In some cases, researchers have observed a reverse shift from an interior crack failure to a surface crack failure mode after specimens were treated with SP, SMAT, or LSP.

Shiozawa and Lu [10] performed rotary bending fatigue tests on SP-treated JIS SUJ2 steel specimens, and the test results showed that SP greatly improved the fatigue properties of the specimens in the HCF regime; however, the reverse effect was observed in the VHCF regime (reference Figure 6). In the research from Myung et al. [29], rotary bending test results showed that the effects of SP on the fatigue properties of SAE 9254 steel specimens went from positive to negative at approximately $N_f = 10^8$ cycles. On the other hand, the study by Trško et al. [12] showed an opposite trend in the improvement of fatigue properties after treating 50CrMo4 spring steel specimens with severe shot peening. Per the uniaxial test results, the level of improvement induced by severe shot peening was instead greater at lower fatigue stress levels. Research from Zhang et al. [13] and Suh et al. [30] showed that SP improved the bending-load fatigue properties of 35CrMo steel, LZ50 steel, and 7075-T651 aluminum specimens in the HCF and VHCF regimes. Nevertheless, the test results from Tian et al. [31] and Suh et al. [30] showed that SP had detrimental effects on the axial-load fatigue properties of 2024-T351 and 7075-T651 aluminum specimens in the HCF and VHCF regimes.

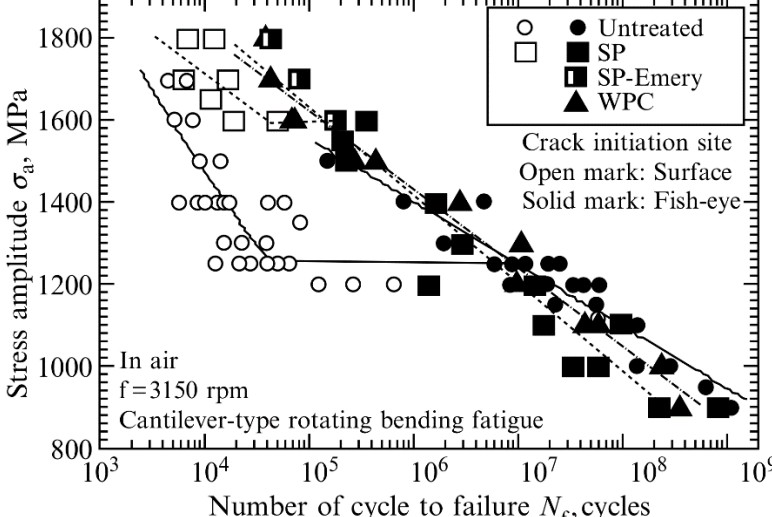

**Figure 6.** Comparison of rotary bending fatigue test results for different batches of JIS SUJ2 specimens. The specimens treated with 500 μm shot peening are marked as SP, and specimens treated with 55 μm fine particle shot peening (also known as the WPC treatment) are marked as WPC. The SP-Emery specimens were fabricated by post-polishing SP specimens. Reprinted with permission from Ref. [10]. Copyright 2002 Wiley.

According to research from Gao et al. [5], SMAT had positive effects on the fatigue lives of 7075-T6 aluminum specimens subjected to a uniaxial fatigue load at a stress level of 220 MPa; however, negative effects were observed at a stress level of 180 MPa (reference Table 3 and Figure 7). The as-received specimens were electropolished and referred to as the EP batch. Specimens from the Steel-2 and Steel-3 batches were subjected to different SMAT treatments, and specimens from the Steel-3MP batch underwent additional post-polishing. At a fatigue stress level of 220 MPa, the specimens from the Steel-2 batch had on average a 193% longer fatigue life than the EP batch; the specimens from the Steel-3 batch had a 154% longer fatigue life than the EP batch (reference Figure 7b). At a fatigue stress level of 180 MPa, the specimens from the Steel-2 batch had on average a 68% shorter fatigue life than the EP batch, and the specimens from the Steel-3 batch had a 95% shorter fatigue life than the EP batch (reference Figure 7c). Additionally, Gao et al. [32] also studied the effects of SMAT on the high-cycle and very-high-cycle fatigue properties of TC11 titanium specimens. Uniaxial ultrasonic test results showed that SMAT negatively affected the fatigue properties of the specimens in the HCF and VHCF regimes, and interior crack failures were no longer observed after the SMAT treatment.

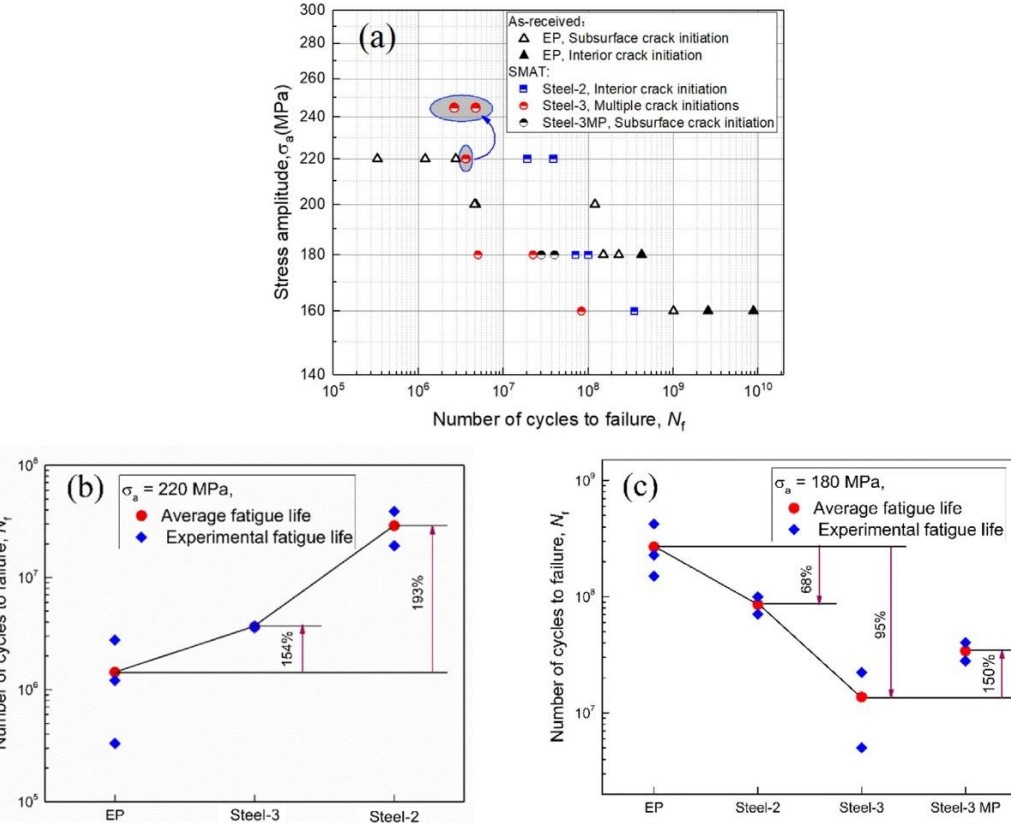

**Figure 7.** (**a**) S-N plot of the uniaxial ultrasonic fatigue test results of 7075-T6 aluminum specimens in as-received (EP) condition and after Steel-2, Steel-3, and Steel-3MP SMAT treatments. (**b**) Comparison of average fatigue lives at $\sigma_a$ = 220 MPa. (**c**) Comparison of average fatigue lives at $\sigma_a$ = 180 MPa. Reprinted with permission from Ref. [5]. Copyright 2020 Elsevier.

**Table 3.** Comparison of treatments, surface roughness, and maximum compressive residual stress of 7075-T6 aluminum specimens in as-received (EP) condition and after Steel-2, Steel-3, and Steel-3MP SMAT treatments [5].

| Specimen Condition | Treatment | Surface Roughness | | Max Compressive Residual Stress (MPa) |
|---|---|---|---|---|
| | | *Ra* (μm) | *Rz* (μm) | |
| EP | Electropolished (as received) | 0.35 | 0.82 | N/A |
| Steel-2 | SP with 2 mm steel shots at 30% power for 25 min | 1.11 | 4.66 | −230 |
| Steel-3 | SP with 3 mm steel shots at 30% power for 15 min + 50% power for 5 min | 1.77 | 7.48 | −330 |
| Steel-3MP | Steel-3 + post-polishing | 0.41 | 1.94 | N/A |

In the research from Qin et al. [33] and Jiang et al. [14], uniaxial ultrasonic test results showed that LSP worsened the high-cycle and very-high-cycle fatigue properties of 2024-T351 aluminum and heat-treated AM Ti-6Al-4V titanium specimens, respectively. However, Yang et al. [34] observed that LSP improved the uniaxial ultrasonic fatigue test results for Ti-8Al-1Mo-1V titanium alloy in the HCF and VHCF regimes. Wang et al. [7] performed three-point bending fatigue tests on six batches of forged Ti-6Al-4V titanium specimens treated with six different LSP configurations. Only one of the batches showed worsening bending fatigue properties in the HCF and VHCF regimes after LSP treatment.

*3.3. Causes for Strengthening and Weakening Effects*

The studies listed above show that surface treatments can have different effects on the high-cycle and very-high-cycle fatigue properties of metals. While SMRT and UNSM have been shown to consistently improve the fatigue properties of treated metals in the HCF and VHCF regimes, the effects of SP, SMAT, and LSP are not always positive. Many researchers have investigated the causes for this noted difference in the effects of surface treatments. It has been found that the effects of surface treatments on the high-cycle and very-high-cycle fatigue properties of metals are mainly caused by treatment-induced changes in the surface finish, microhardness and microstructure, and residual stresses of metals. Additionally, based on the crack initiation mechanisms presented in the introduction section, the combined effects of these treatment-induced changes can also cause the failure mode to shift between surface crack and interior crack failures. The following sections will review these treatment-induced changes and discuss how the high-cycle and very-high-cycle fatigue properties of metals are strengthened or weakened with respect to the aspects of surface finish, microhardness and microstructure, and residual stresses.

**4. Surface Finish**

Surface finish is one of the main factors for fatigue crack initiation on material surfaces. The material surface roughness and surface defects affect the SIF on the material surface. For a given fatigue load, a specimen with a coarse surface will have a higher surface SIF range, $\Delta K_s$, than the same specimen with a smooth surface. Additionally, surface defects also serve as stress concentration areas that promote crack initiation. When $\Delta K_s$ exceeds the surface threshold SIF range, $\Delta K_{th,s}$, surface cracks will initiate and grow. Thus, specimens with a coarse surface and surface defects are more susceptible to surface cracks. On the other hand, the fatigue properties of metals can generally be improved by reducing surface defects and lowering surface roughness to prevent the initiation of surface cracks. When the surface cracks are suppressed, interior crack failures are more likely to occur [2,10].

Among the surface treatment methods discussed in this paper, the SMRT and UNSM treatments can generally produce a low surface roughness on treated materials [18,35], which contributes to the consistent positive treatment effects on the high-cycle and very-high-cycle fatigue properties of metals. This relationship between the surface treatment effects on the material surface roughness and fatigue properties was exhibited in the re-

search by Dong et al. [1]. Recalling that Dong et al. [1] compared the fatigue test results of SMRT-treated 7075-T651 FSW specimens, and the SMRT + AFT batch had additional microgrooves after the SMRT treatment to simulate the surface condition of as-welded specimens. The fatigue test data showed that the SMRT batch had a higher fatigue strength than the SMRT + AFT batch in the HCF regime before the knee point on the S-N curve (reference Figure 5). Additionally, all fractures initiated from surface crack sites before the knee point. These findings concur with the understanding that the smooth surface condition is beneficial to material fatigue strength, and lower surface roughness reduces the stress concentration on the material surface and helps to suppress the initiation of surface cracks. It is also worth noting that all the fractures initiated from internal crack sites after the knee point for 7075-T651 FSW specimens in SMRT and SMRT + AFT conditions, and there was no obvious fatigue strength difference between the two batches in the interior crack failure mode region. This is mainly because the interior crack initiation and propagation are not directly affected by material surface finish conditions. In contrast, all fatigue fractures were initiated from surface crack sites in the research by Karimbaev et al. [28]. It was indicated by Karimbaev et al. [28] that surface roughness did affect the fatigue properties of SP and UNSM-treated AISI 4340 steel specimens in the VHCF regime. Karimbaev et al. [28] compared the surface roughness of specimens from as-received, SP, UNSM, and SP + UNSM batches. The measured surface roughness values (Table 2) indicated that the UNSM and SP + UNSM batches had improved surface roughness compared to the SP and as-received batches; the SEM images of the surface microstructure from each batch are shown in Figure 8. Since only surface crack originated failures were observed in all specimens, and knowing that a smooth surface suppresses the initiation of surface cracks, it was logical for Karimbaev et al. [28] to note that improved surface roughness contributed to better fatigue properties for the UNSM and SP + UNSM batches compared to the SP and as-received batches (Table 2). Furthermore, research by Sakai et al. [4] indicated that additional polishing after surface rolling treatment tends to give JIS SUJ2 steel specimens longer fatigue lives than specimens only treated with surface rolling treatment in the HCF regime. Sakai et al. [4] noted that post-treatment polishing could still be beneficial to material fatigue properties, and it barely affected the compressive residual stress in the surface region. This observation highlighted the importance of having a smooth surface and suggested a method to further improve material fatigue properties after SMRT treatment.

When compared to the SMRT and UNSM treatments, LSP was observed to have different effects on the surface roughness of metals in other studies. The research from Qin et al. [33] and Jiang et al. [14] showed that increasing the LSP intensity increased the surface roughness of LSP-treated 2024-T351 aluminum and additive manufactured (AM) Ti-6Al-4V titanium specimens. On the contrary, Nalla et al. [36] reported that LSP had no obvious effects on the surface roughness of Ti-6Al-4V specimens compared to a significant improvement from surface rolling treatment. Thus, the effects of LSP-induced changes on surface roughness are usually reviewed on a case-by-case basis.

Rather than improving surface conditions, the surface roughness of a metal generally increases after shot peening and SMAT treatments. The impacts of the shots during SP and SMAT treatments can create dimples that roughen metal surfaces, and the coarse surface condition negatively affects the fatigue properties of the metals. For instance, Gao et al. [5] polished two specimens in a batch of four SMAT-treated 7075-T6 aluminum specimens and subjected them to ultrasonic fatigue tests. At a fatigue stress amplitude of 180 MPa, the polished specimens (Steel-3MP) had on average a 150% longer fatigue life (reference Figure 7c). In addition to increasing surface roughness, surface defects can be induced by SP and SMAT treatments. In the research by Trško et al. [12], severe plastic deformation was observed in the surface region of severe shot peening-treated 50CrMo4 steel specimens. This severe plastic deformation induced sharp notches and peeling on the surface of the specimens, as shown in Figure 9. Similarly, Myung et al. [29] observed that micro-pits were occasionally generated on the shot peening-treated surface of SAE 9254 specimens,

but not on the untreated surface. The defects noted above were detrimental to the fatigue properties of the treated materials. They served as stress concentration areas and promoted surface crack initiation. In summary, the SP and SMAT processes are detrimental to the surface finish of metals; thus, it is important to control the treatment intensities to minimize the negative impact on fatigue properties.

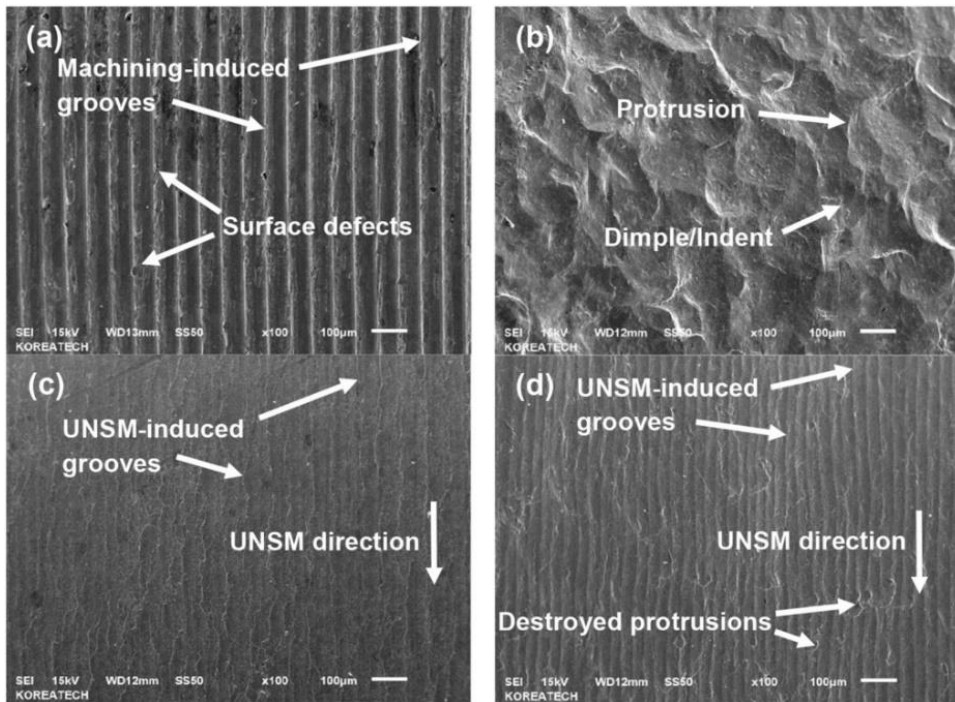

**Figure 8.** Scanning electron microscopy (SEM) images of the surface conditions of AISI 4340 specimens in (**a**) as-received condition and after (**b**) SP, (**c**) UNSM, and (**d**) SP + UNSM treatments. Reprinted with permission from Ref. [28]. Copyright 2020 Elsevier.

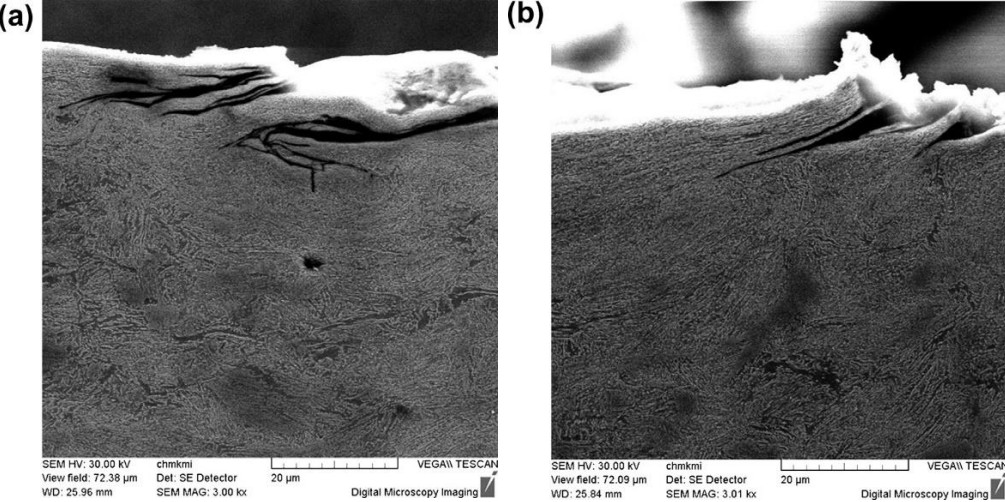

**Figure 9.** Damages in the surface layers of severe shot peening-treated 50CrMo4 steel specimens: (**a**) material peeling off from the surface; (**b**) a sharp notch under the peeled off material layer Reprinted with permission from Ref. [12]. Copyright 2014 Elsevier.

## 5. Microstructure and Microhardness

Cold working surface treatments can modify the microstructure and increase microhardness in the surface region of metals, and these changes can improve the metal fatigue properties in the HCF and VHCF regimes. As mentioned in earlier paragraphs, when metal materials are treated with surface treatment methods, such as SP, LSP, SMRT, SMAT, and UNSM, plastic deformation occurs in the metal surface region. These treatment-induced plastic deformation changes can strengthen the material fatigue properties by refining the surface grain size, increasing the surface microhardness, and forming a nanograin layer and gradient microstructure.

### 5.1. Work-Hardened Layer

Based on the Hall–Petch relation, the material microhardness is increased in the surface region because of the grain refinement induced by surface treatments. The surface microhardness measurement is normally used to evaluate the level of work hardening and detect the thickness of the work-hardened layer. An increase in microhardness will boost the surface resistance to plastic deformation; hence, a crack is less likely to initiate in the surface work-hardened layer.

In the research by Tian et al. [31], SP induced a work-hardened layer of approximately 100 μm in the surface of a 2024-T351 aluminum specimen. Myung et al. [29], Trško et al. [12], Suh et al. [30], and Shiozwa and Lu [10] also observed that material surface microhardness increased after performing SP on SAE 9254 steel, 50CrMo4 steel, 7075-T651 aluminum, and JIS SUJ2 steel specimens, respectively. The research from Gao et al. [32] and Li et al. [17] showed an increase in microhardness after performing SMAT on TC11 titanium and heat-treated AM Ti-6Al-4V titanium specimens. In another study from Gao et al. [5], work hardening affected the material as deep as approximately 600 μm in 7075-T6 aluminum specimens after SMAT treatment, and interior cracks initiated between 560 and 680 μm with small scattering as the result. Qin et al. [33] performed LSP on 2024-T351 aluminum specimens and observed that the maximum microhardness increased by 9.6% and 16.3% with respect to the 10 J and 20 J laser energy levels. According to the studies by Dong et al. [1], a maximum increase of 43.3% in microhardness was observed in FSW 7075-T651 aluminum specimens, and the work-hardened layer reached approximately 550 μm in depth. Similarly, Sakai et al. [4] and Saalfeld et al. [19] observed work-hardened layers under the rolling-treated surface of JIS SUJ2 and SAE 1045 specimens, respectively. The work-hardened layers were able to restrain the internal crack growth, which resulted in an elliptical instead of a circular fisheye area forming around the internal cracks (reference Figure 10). The research from Khan et al. [25], Karimbaev et al. [28], Cao et al. [27], and He et al. [26] showed that UNSM increased the surface microhardness in AISI 310 stainless steel, AISI 4340 steel, Ti-6Al-4V titanium, and welded SMA490BW specimens, respectively. As shown by the above studies, cold working surface treatments are effective in inducing a higher microhardness of metals. Because SMRT and UNSM also tend to produce a smooth surface finish, inducing the work-hardened layer via these two treatment methods can maximize the improvement of the fatigue properties of metals.

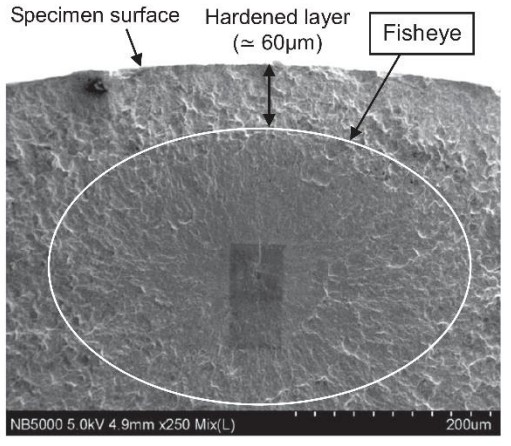 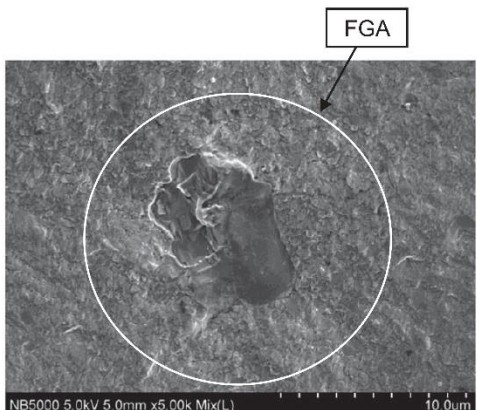

(a) Fish-eye formed on the fracture surface     (b) Inclusion and FGA at crack initiation site

**Figure 10.** SEM observations of the fracture surface of the interior inclusion-induced fracture from rolling-treated JIS SUJ2 specimens: (**a**) fisheye formed on the fracture surface; (**b**) inclusion and FGA at the crack initiation site. Reprinted with permission from Ref. [4]. Copyright 2015 Wiley.

### 5.2. Plastic Deformation Zone, Nanograin Layer, and Gradient Microstructure

In addition to the work-hardened layer, the subsurface plastic deformation zone is another indication of the strengthened microstructure induced by surface treatments. Grain refinement is generally observed in the plastic deformation zone, and it can further induce the nanograin layer and gradient microstructure to greatly enhance the fatigue properties of treated metals. The typical nanocrystalline grain refinement process is shown in Figure 11 [7,23,28]. During the surface treatment process, the material undergoes plastic deformation due to the high strain rate and high-density dislocations occur, forming dislocation walls and dislocation entanglements. As plastic deformation continues, subgrain boundaries, dislocation cells, and/or deformation twins are gradually formed due to dislocation motions, such as gliding, interaction, accumulation, tangling, rearrangement, and annihilation. The subgrains then rotate with further increases in plastic strain and strain rates, which eventually leads to subgrain boundaries with a greater orientation and transforms low-angle grain boundaries (LAGBs) into high-angle grain boundaries (HAGBs). As the subgrain and HAGBs continue to form, the grain size is then gradually refined down to the nanoscale. Thus, the gradient microstructure may also be induced during the surface treatment process if the plastic strain and strain rate also exhibit a gradient distribution along the depth direction [7,23,28].

SP can induce a plastic deformation zone in the surface region of metals; however, the standard SP treatment tends to encounter difficulties in inducing the nanograin layer and gradient microstructure due to milder plastic strain. Myung et al. [29] observed a 20 μm-thick plastic deformation zone under the surface of SP-treated SAE 9254 steel specimens, and research from Suh et al. [30] showed that SP induced a 100 μm thick plastic deformation zone in the surface region of 7075-T651 aluminum specimens. Neither Myung et al. [29] nor Suh et al. [30] reported any observation of the nanograin layer and microstructure. On the other hand, Trško et al. [12] observed a 20 μm-thick nanograin layer after the severe shot peening treatment of 50CrMo4 steel specimens.

SMAT and LSP treatments can generally induce more severe plastic deformation than standard SP; thus, the nanograin layer is often observed and a gradient microstructure may also form. According to research from Gao et al. [32], the nanograin layer was observed after treating TC11 specimens with SMAT, and there was a transition region between the nanograin layer and the bulk material grain layer. Kumar et al. [37] also found that a similar nanograin layer formed below the ultrasonic shot peening-treated surface of Ti-6Al-4V specimens. Per the study by Qin et al. [33], a very thin nanograin layer was formed under the surface treated by LSP in 2024-T351 aluminum specimens, and there was no major

change in grain size in the region right below the nanograin layer. Yang et al. [34] observed that LSP induced plastic deformation and increased the number of deformation twins and HAGBs under the treated surface of Ti-8Al-1Mo-1V titanium specimens; however, the grain size was not refined. To the contrary, in the research by Jiang et al. [14], grains with sizes less than 2 μm was found in the topmost layer of LSP-treated AM Ti-6Al-4V titanium specimens, and the deformation zone was measured to be up to 75 μm in depth. The study by Wang et al. [7] also reported a nanograin layer of less than 1 μm that was formed under the surface of LSP-treated Ti-6Al-4V titanium specimens, and twining was the main cause of grain refinement. A dislocation cell layer was observed below the nanograin layer, followed by the high-density dislocation layer, and the original coarse grain layer (reference Figure 12).

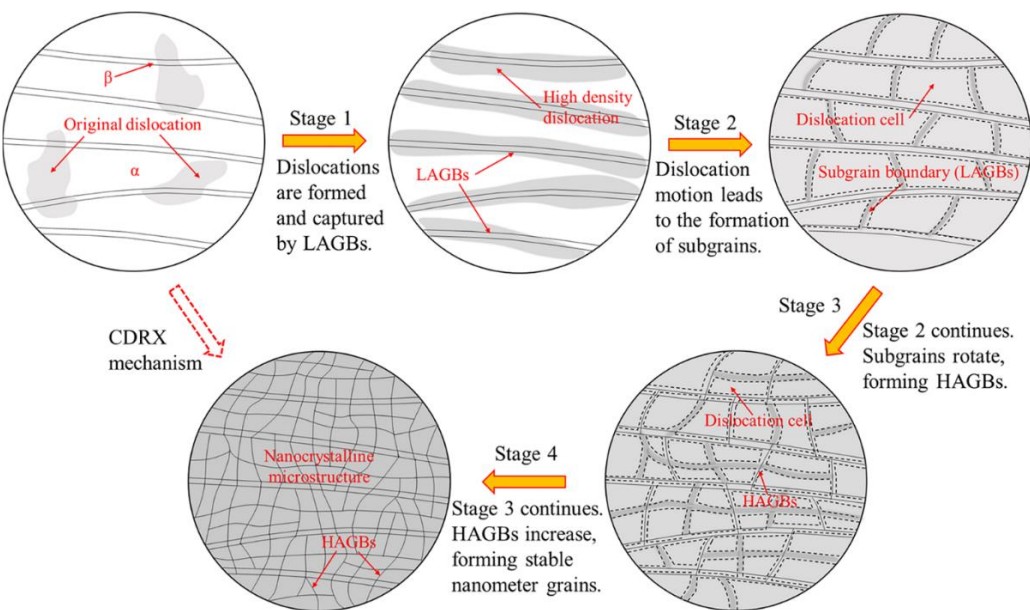

**Figure 11.** The formation process of surface nanocrystalline microstructure. Reprinted with permission from Ref. [7]. Copyright 2022 Elsevier.

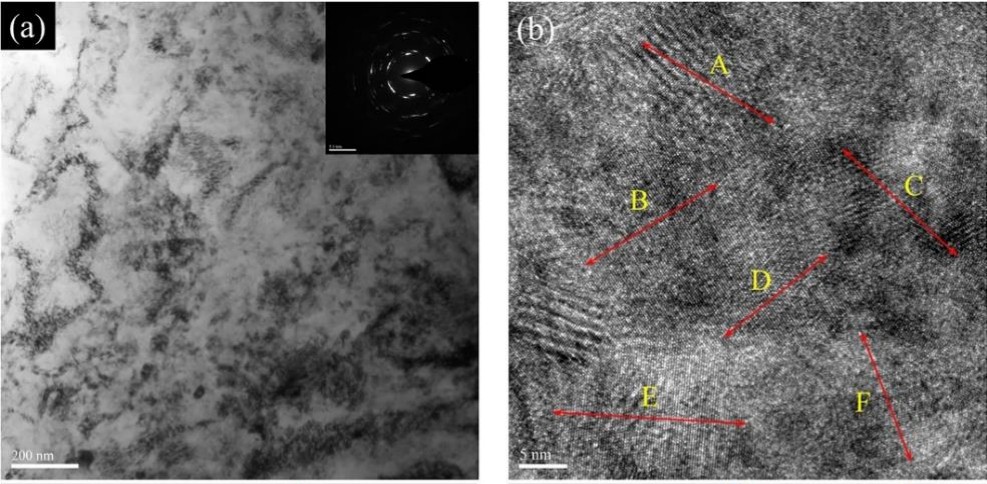

**Figure 12.** *Cont*.

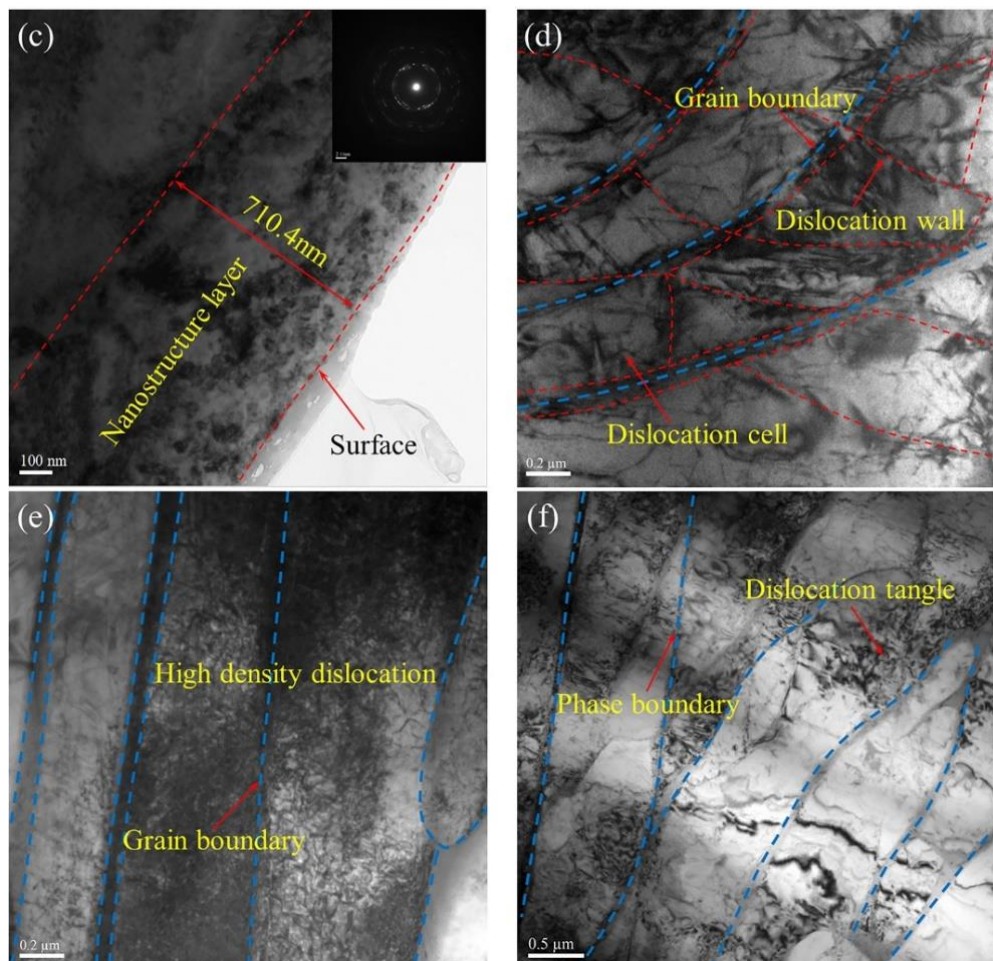

**Figure 12.** Transmission electron microscopy (TEM) morphology of the microstructure in the surface region of the LSP-treated Ti-6Al-4V titanium specimen: (**a**) surface nanocrystals; (**b**) high-resolution image of (**a**) showing different nanocrystal orientation; (**c**) nanocrystal layer; (**d**) dislocation cells; (**e**) high-density dislocations; (**f**) low-density dislocations. Reprinted with permission from Ref. [7]. Copyright 2022 Elsevier.

SMRT and UNSM are effective in inducing a severe plastic deformation zone, and many researchers have studied the positive effects associated with the nanograin layer and gradient microstructure on the high-cycle and very-high-cycle fatigue properties of metals treated with SMRT and UNSM. It is generally more difficult to induce a gradient microstructure on aluminum; however, Liu et al. [38] observed a ~500 μm-thick gradient microstructure layer that formed on 7075-T651 aluminum specimens after SMRT treatment, and noted that the gradient microstructure strengthened the high-cycle fatigue strength of the specimens. In the research by Dong et al. [1], a gradient microstructure was observed in the surface region of SMRT-treated FSW 7075-T651 aluminum specimens. The grain size increased from the surface to the interior forming the nanograin layer (NGL), deformed grain layer (DGL), and coarse grain layer (CGL). The average grain size in the nanograin layer was refined to approximately 250 nm, and the amount of low-angle grain boundaries decreased after SMRT treatment. Dong et al. [1] noted that the gradient microstructure can significantly strengthen the fatigue properties of FSW 7075-T651 aluminum specimens by preventing crack initiation at the surface and delaying crack growth in the interior (reference Figure 13). In terms of UNSM, He et al. [39] investigated microstructure changes after FSW 7075-T651 aluminum specimens were processed with ultrasonic peening treatment. A severely deformed layer was observed under the treated surface with a thickness of approximately 120 μm, and the sizes of Fe-rich inclusions were refined



in the deformation zone. This effect lowers the stress concentration at the inclusion sites. Furthermore, Cao et al. [27] and Karimbaev et al. [28] found that UNSM treatment induced severely plastically deformed layers on Ti-6Al-4V titanium and AISI 4340 steel specimens, respectively. He et al. [26] investigated changes in the microstructure of four batches of welded SMA490BW steel specimens treated with different ultrasonic impact treatment parameters, and the gradient microstructure became more obvious as the thickness of the plastic deformation zone increased. Although not as common as in workpieces treated with SMRT and UNSM, some researchers also observed a clear gradient microstructure on other selected titanium and steel workpieces after treatment with the appropriate SP, SMAT, and LSP methods [40–43]. Nevertheless, these studies did not investigate the effects of SP, SMAT, and LSP-induced gradient microstructure on the very-high-cycle fatigue properties of the selected metals; therefore, these may still be potential areas for future studies.

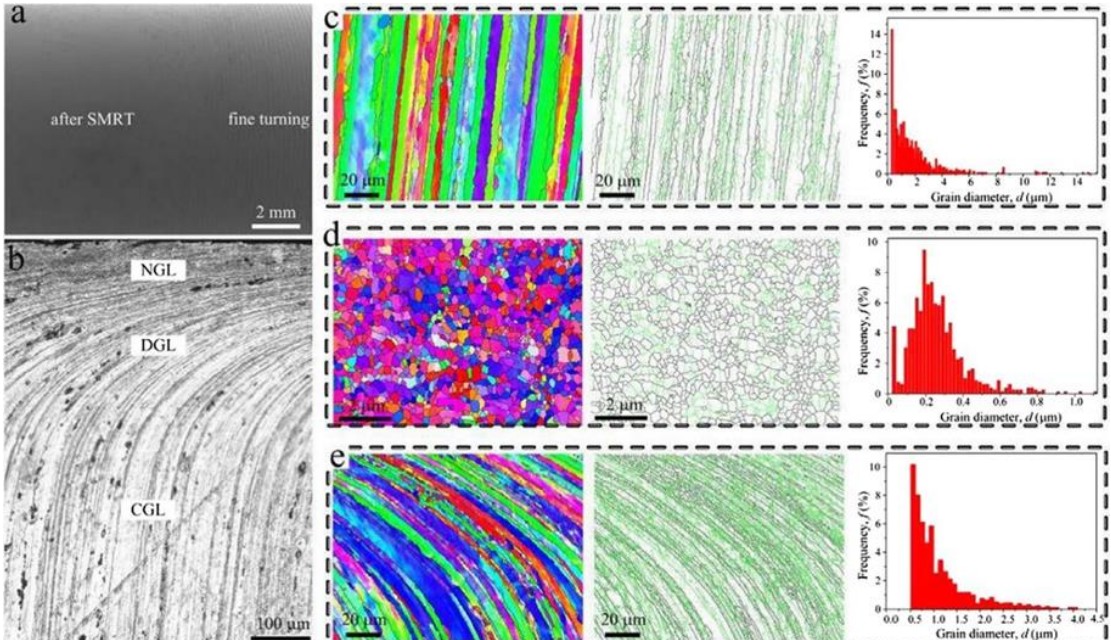

**Figure 13.** The surface morphology of SMRT-treated FSW 7075-T651 aluminum specimens: (**a**) surface with and without microgrooves induced by post-SMRT fine turnings; (**b**) cross-sectional microstructure of the gauge consisting of NGL, DGL, and CGL; (**c**–**e**) electron backscatter diffraction (EBSD) results of CGL, NGL, and DGL, respectively. Reprinted with permission from Ref. [1]. Copyright 2019 Elsevier.

The deformation zone, nanograin layer, and gradient microstructure can strengthen the fatigue properties of metals in several ways. Firstly, the surface hardness is significantly increased with the formation of the nanograin layer based on the Hall–Petch relation. Due to grain refinement, the size of weak facets, inclusions, and other defects are also refined in the deformation zone, which lowers the stress concentration in those areas. The smaller grain size also reduces the slip distances of the slip bands; thus, the stress concentration is lower at the grain boundary because of less dislocation accumulation [7]. Crack initiation is then suppressed as the stress concentration areas are slowed and reduced. Additionally, the refined grain size also makes it harder for further plastic deformation to occur because dislocation movements are blocked by more grain boundaries, which increases the resistance to crack initiation [7]. Similarly, the growth of microcracks is slowed by additional grain boundaries due to refined grains, and this is important for the very-high-cycle fatigue properties of metals because more microcracks tend to form at lower stress levels. On the other hand, grain refinement can promote crack propagation by causing the crack growth threshold to lower. Therefore, gradient microstructure becomes

the optimized option because the surface nanograin layer can prevent crack initiation, and the coarser interior can slow crack propagation.

## 6. Residual Stress

Residual stress is an important part of the effects of cold working surface treatment on the fatigue properties of metals in the HCF and VHCF regimes. Compressive residual stresses are induced in the metal surface region due to material plastic deformation caused by the surface treatments. Increasing compressive residual stress can improve metal fatigue properties. However, obtaining a high level of compressive residual stress may come with sacrifices in surface finish, and the level of compressive residual stresses can relax throughout the metal fatigue process. Furthermore, the tensile residual stresses are also induced by the surface treatment process, and can be detrimental to the fatigue properties of metals. This section will discuss these key aspects of residual stresses regarding the effects of different surface treatments.

### 6.1. Compressive Residual Stress

Compressive residual stresses in the metal surface region can have beneficial effects on the high-cycle and very-high-cycle fatigue properties of metals. The compressive residual stresses, which are induced by surface treatment processes, are generally concentrated within a compressive zone right below the treated surfaces. When a fatigue load is applied to the surface of treated metals, the local mean stress is reduced in the compressive zone due to the superposition of the applied stress and residual stress. Therefore, increasing the compressive residual stress can lower the local SIF range, $\Delta K$, which helps to suppress crack initiation and slow crack propagation in the compressive zone. Subsequently, these effects improve metal fatigue properties in the HCF and VHCF regime by delaying failures from surface crack fractures and letting internal crack failures occur at higher stress levels. For instance, after treating FSW 7075-T651 specimens with SMRT, Dong et al. [1] noted that the induced compressive residual stress had positive effects on the fatigue properties of the SMRT-treated specimens in the HCF and VHCF regimes, and interior-crack failures in the VHCF regime were only observed after SMRT treatment (reference Figure 5). This observation shows that treatment-induced compressive residual stress suppresses surface crack fractures. Similarly, in research by Suh et al. [24], it was observed that increasing the static load during UNSM treatment will induce greater compressive residual stresses in the surface region of JIS SCM435 steel specimens (reference Table 1). Furthermore, the fatigue properties of the specimens in the HCF and VHCF regimes also improved as the compressive residual stress level increased, which suggests a positive correlation between the two measurements. As shown in these two references, the compressive residual stress is a contributing factor for the beneficial effects of surface treatments on the fatigue properties of metals. For all surface treatment methods discussed in this paper, the level of compressive residual stress can be maximized by adjusting the surface treatment configurations. However, maximizing the compressive residual stress might not always improve the fatigue properties of treated specimens.

### 6.2. Correlation with Surface Roughness

Increasing the level of compressive residual stress through surface treatments can come at a cost of deteriorating surface finish. Since compressive residual stress is induced through metal plastic deformation during the surface treatment process, an increase in the severity of metal plastic deformation is required to induce greater compressive residual stress. This can generally be achieved by adjusting the configurations of the treatment processes. However, for treatment methods such as shot peening and SMAT, the surface finish in the treated area can significantly worsen as the severity of plastic deformation increases. Research by Gao et al. [5] showed that inducing more compressive residual stress with SMAT worsened the surface roughness of 7075-T6 aluminum specimens (reference Table 3). As the compressive residual stress increased and surface roughness worsened,

the fatigue lives of the treated specimens also deteriorated (reference Figure 7), and the fracture initiating sites shifted from internal cracks to multiple crack sites at both the interior and surface. This observation suggests that the beneficial effects of having a higher compressive residual stress were outweighed by the negative effects due to deteriorating surface roughness. Similarly, Shiozawa and Lu [10] observed that the surface roughness of JIS SUJ2 steel specimens deteriorated as SP treatment induced more compressive residual stress in the metal surface region (reference Figures 6 and 14). As the compressive residual stress increased and surface roughness worsened, the fatigue failure mode shifted from interior crack to surface crack failure in the higher fatigue stress regime, which essentially reduced the fatigue lives of the specimens at those fatigue stress levels. Nevertheless, Shiozawa and Lu [10] found that additional polishing after SP treatment could reinstate the interior crack failure mode and further enhance the fatigue lives of SP-treated specimens (reference Figure 6). On the other hand, SP-treated JIS SUJ2 steel specimens had inferior fatigue properties compared to untreated specimens in the VHCF regime. This condition was not caused by the coarse surface roughness on SP-treated specimens, but rather by SP-induced tensile residual stress due to residual stress distribution.

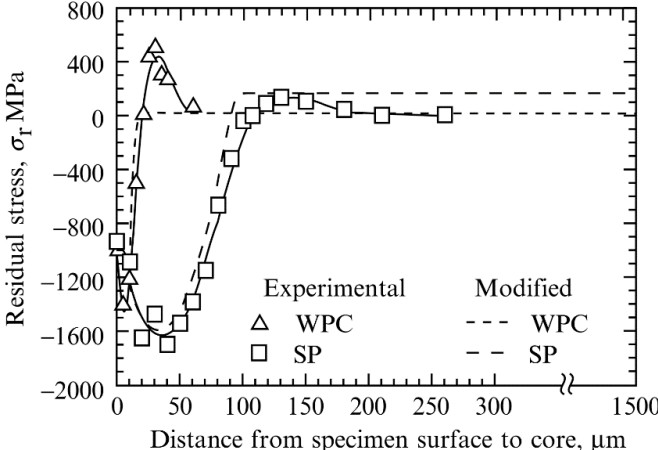

**Figure 14.** Experimental and modified results of residual stress distribution in SP-treated JIS SUJ2 steel specimens. The modified residual stress distribution shown by dashed lines reflects the stress redistribution after material surfaces were electropolished during the measurements. The specimens treated with 500 μm and 55 μm shots are noted as SP and WPC, respectively [10]. Reprinted with permission from Ref. [10]. Copyright 2002 Wiley.

*6.3. Tensile Residual Stress*

Tensile residual stress is another adverse factor that can be induced by the surface treatment methods discussed in this paper. Tensile residual stress always coexists with compressive residual stress to satisfy the equilibrium condition. Whenever a surface treatment induces compressive residual stress in the surface region of a metal workpiece, the tensile residual stress is subsequently induced in the metal core region due to the equilibrium of residual stress distribution. Contrary to compressive residual stress, the tensile residual stress causes the mean stress to increase, which promotes the initiation of internal cracks and accelerates internal crack propagation. As discussed in the paragraph above, Shiozawa and Lu [10] observed that increasing the compressive residual stress through SP treatment raised the tensile residual stress in the core region of JIS SUJ2 steel specimens, which caused interior crack failures to occur earlier (reference Figure 6). The study by Qin et al. [33] showed that tensile residual stress was a contributing factor that worsened the fatigue properties of LSP-treated 2024-T351 aluminum specimens as the LSP intensity increased. Additionally, Saalfeld et al. [19] observed that increasing the compressive residual stress strengthened the fatigue properties of surface rolling-treated SAE 1045 steel specimens in the fatigue regime dominated by surface crack failures.

However, the coexisting tensile residual stress weakened the fatigue properties of surface rolling-treated SAE 1045 steel specimens in the fatigue regime dominated by internal crack failures. Overall, because internal crack failures tend to occur in the VHCF regime, the induced tensile residual stress generally affects the fatigue properties of metal negatively in the VHCF regime.

### 6.4. Stress Relaxation

Finally, the relaxation of residual stress can gradually diminish the effectiveness of the induced residual stress. The study by Dong et al. [1] showed that more and faster stress relaxation can cause surface crack failures to occur at lower fatigue stress levels in SMRT-treated FSW 7075-T651 specimens. Research by He et al. [39] indicated that the residual stress on UIT-treated FSW 7075-T651 specimens relaxed the most in the low-cycle fatigue (LCF) regime and then relaxed at much slower rates in the HCF and VHCF regimes. Zhang et al. [13] found that the relaxation curves of microshot-peened 35CrMo and LZ50 steel specimens were affected by the residual stress distribution, fatigue load distribution, and material cyclic yield strengths. Zhang et al. [13] also showed that increasing the applied fatigue load could accelerate and increase residual stress relaxation in microshot-peened 35CrMo and LZ50 steel specimens. The cyclic yield strengths for 35CrMo and LZ50 steel specimens were 705 MPa and 313 MPa, respectively. Figure 15 shows the relaxation curves of the compressive residual stress in microshot-peened 35CrMo steel specimens at fatigue stress levels of 600 MPa and 650 MPa, and the relaxation curve of the compressive residual stress in microshot-peened LZ50 steel specimens at a fatigue stress level of 260 MPa. These fatigue stress levels were lower than the material cyclic yield strengths; thus, the compressive residual stresses quickly relaxed in the first few cycles and then became relatively stable. Figure 16 shows the relaxation curves of compressive residual stress in microshot-peened LZ50 steel specimens at fatigue stress levels of 380 MPa and 360 MPa. These fatigue stress levels were higher than the material yield strength; thus, the compressive residual stresses for the LZ50 specimens continued to relax until approximately $N_f = 10^4$ cycles before stabilizing. It is worth noting that the compressive residual stress was almost completely relaxed at the fatigue stress level of 380 MPa. Since the subsurface compressive residual stress tends to improve metal fatigue properties in the HCF regime and the internal tensile stress tends to worsen the fatigue properties of metals in the VHCF regime, it will be beneficial to optimize the residual stress relaxation curve such that the metal can benefit the most from residual stress throughout the fatigue life.

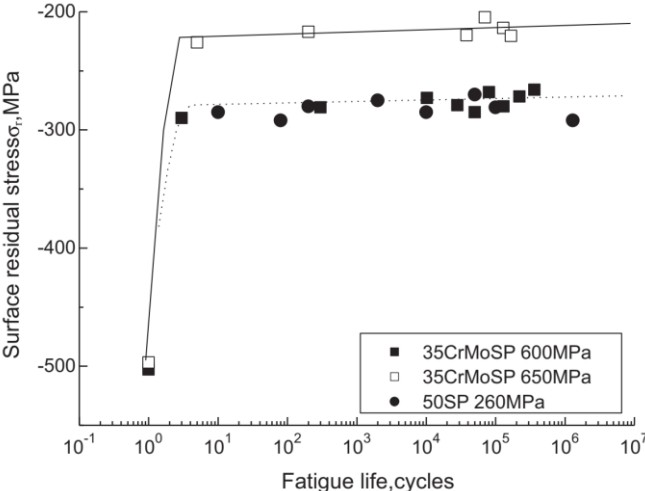

**Figure 15.** Comparison of compressive residual stress relaxation curves of microshot-peened 35CrMo and LZ50 steel specimens at relatively low fatigue stress levels compared to the material cyclic yield strengths. The specimens were subjected to rotary bending fatigue tests. Reprinted with permission from Ref. [13]. Copyright 2011 Elsevier.

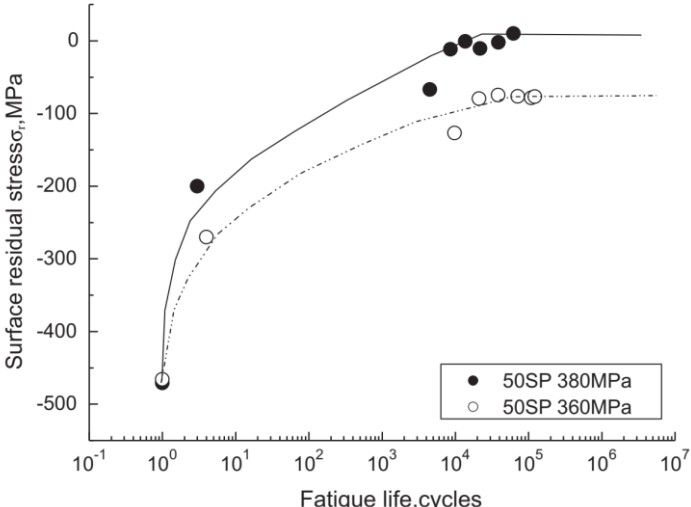

**Figure 16.** Comparison of compressive residual stress relaxation curves of microshot-peened 35CrMo and LZ50 steel specimens at relatively high fatigue stress levels compared to the material cyclic yield strengths. The specimens were subjected to rotary bending fatigue tests. Reprinted with permission from Ref. [13]. Copyright 2011 Elsevier.

## 7. Conclusions

In this paper, SP, SMAT, SMRT, LSP, and UNSM were compared and reviewed based on the treatment effects on the fatigue properties of metals in the HCF and VHCF regimes. The main conclusions are as follows:

1.  SP, SMAT, and LSP do not have consistent positive effects on the fatigue properties of metals in the HCF and VHCF regimes. Nevertheless, SMRT and UNSM treatments generally improve metal fatigue properties in the HCF and VHCF regimes.
2.  Improving the surface finish condition of metals can improve the fatigue properties in the HCF and VHCF regimes because a smooth surface finish helps to prevent surface crack initiation. SMRT and UNSM treatments tend to produce a smooth surface finish on metals, whereas SP and SMAT tend to worsen the surface finish on metals. Furthermore, the effects of LSP on the surface finish of metals varied in different studies.
3.  All surface treatment methods discussed in this paper can induce a plastic deformation zone and increase microhardness in the surface region of metal. LSP, SMRT, and UNSM can induce plastic deformation with sufficient plastic strain and strain rate to form a nanograin layer and gradient microstructure. These microstructure features can prevent surface crack initiation and delay crack propagation, which essentially improves metal fatigue properties in the HCF and VHCF regimes.
4.  All surface treatment methods discussed in this paper can induce compressive residual stress in the surface region of treated metals, and tensile residual stress will coexist in the metal interior region to satisfy the equilibrium of residual stress distribution. Having compressive residual stress can improve metal fatigue properties in the HCF regime because it helps to prevent initiation and delay the propagation of surface cracks. Nevertheless, increasing the level of compressive residual stress through shot peening and ultrasonic shot peening methods will worsen the metal surface finish, which may void the positive effects of having compressive residual stress. Furthermore, the internal tensile residual stress tends to worsen the fatigue properties of metals in the VHCF regime because it generally promotes internal crack initiation and accelerates crack propagation. Stress relaxation occurs during the fatigue process, and the relaxation curve is influenced by the fatigue stress level, residual stress distribution, and material yield strength.

5. There are other surface treatment methods, such as surface mechanical grinding treatment (SMGT) and ultrasonic surface rolling process (USRP), which have limited studies available on their effects on the very-high-cycle fatigue properties of metals. It will be beneficial to also compare these treatments with the other surface treatments noted in this paper.

**Author Contributions:** Writing—original draft preparation, R.C.; supervision, H.X.; writing—review and editing, H.X. and B.L. All authors have read and agreed to the published version of the manuscript.

**Funding:** National Natural Science Foundation of China (91860206).

**Data Availability Statement:** Data sharing not applicable.

**Conflicts of Interest:** The authors declare no conflict of interest.

## Abbreviations

| Abbreviation | Explanation |
| --- | --- |
| AISI | American Iron and Steel Institute |
| AM | Additive manufacture |
| CGL | Coarse grain layer |
| DGL | Deformed grain layer |
| DR | Deep rolling |
| EBSD | Electron backscatter diffraction |
| FGA | Fine granular area |
| FSW | Friction-stir weld |
| HAGB | High-angle grain boundary |
| HCF | High-cycle fatigue |
| JIS | Japanese Industrial Standards |
| LAGB | Low-angle grain boundary |
| LCF | Low-cycle fatigue |
| LSP | Laser shock peening |
| NGL | Nanograin layer |
| SAE | Society of Automotive Engineers |
| SEM | Scanning electron microscopy |
| SIF | Stress intensity factor |
| SMAT | Surface mechanical attrition treatment |
| SMGT | Surface mechanical grinding treatment |
| SMRT | Surface mechanical rolling treatment |
| SP | Shot peening |
| TEM | Transmission electron microscopy |
| UIT | Ultrasonic impact treatment |
| UNSM | Ultrasonic nanocrystal surface modification |
| USRP | Ultrasonic surface rolling process |
| USSP | Ultrasonic shot peening |
| VHCF | Very-high-cycle fatigue |
| WRA | White rough area |

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
