# Peer review of "Comparison of SP, SMAT, SMRT, LSP, and UNSM Based on Treatment Effects on the Fatigue Properties of Metals in the HCF and VHCF Regimes"

_metals, doi:10.3390/met12040642_

Round 1

Reviewer 1 Report

Some corrections/improvements are required:

  1. Lines 258-259: “When the surface SIF range, …
    Something is missing in this part of the sentence.

  2. SMGT
    The abbreviation SMGT appears for the first time in the line 121, but the explanation can be first found in lines 627–628. Make sure that the abbreviation is explained where it appears for the first time.

  3. Explanations of the following abbreviations are missing:    
  • AFT (Fig. 5 and lines 173, 271 and 273)
  • LAP (line 408)
  • HABG (line 418)
  • WPC (Fig. 6 and Fig. 15)
  • LCF (line 573)

4. Figure numbers

  • Figures on pages 10, 11, 12, and 13 have incorrect numbers.
  • Page 10, line 303: Fig 2 - the correct number is Fig. 8
  • Page 11, line 332: Fig.3 - the correct number is Fig. 9
  • Page 12, line 376: Fig. 4 - the correct number is Fig. 10
  • Page. 13, line 398: Fig. 5 - the correct number is Fig. 11

5. Figure captions

If a figure shows microstructures or any other data gained by researching a specific material and a specific surface treatment, the figure caption should specify the materials designation and the applied surface treatment. This was not the case at Fig. 5, Fig. 3 (page 11, the correct number would be Fig. 9), Fig.4 (page 12, the correct number would be Fig. 10), Fig. 12, Fig. 13, and Fig. 14.

6. Following figures are not mentioned in the text:

  • Fig. 1, Fig. 2, Fig. 3 and Fig. 4.

Author Response

Comment 1: Line 258-259: “When the surface SIF range, …”. Something is missing in this part of the sentence.

Response: The sentence was “When the surface SIF range, ΔKs exceeds the surface threshold SIF range, ΔKth,s, surface cracks will initiate and grow…”. It’s now revised as “When ΔKs exceeds the surface threshold SIF range, ΔKth,s, surface cracks will initiate and grow…”. It’s noted that the ΔKs meaning was already explained in an earlier sentence, so it doesn’t need to be repeated again here. We hope this change will make the sentence easier to understand.

Comment 2: The abbreviation SMGT appears for the first time in the line 121, but the explanation can be first found in lines 627-628. Make sure that the abbreviation is explained where it appears for the first time.

Response: Line 121 should be revised. It was “SMRT or Deep Rolling is a surface treatment method very similar to SMGT …”. It’s now revised as “SMRT is the surface rolling treatment method that is also known as DR…” We also noticed that abbreviation deep rolling was wrongly typed as “DP” at two places, should be “DR”. They are now corrected as well.

Comment 3: Explanations of the following abbreviations are missing:

1) AFT (Fig. 4 and lines 173, 271, and 273); Response: the abbreviation SMRT+AFT was first shown in Figure 5 and its caption. We have revised the caption text and added a sentence to better clarify the abbreviation. The specimens with additional microgroove added after processing with SMRT are referred to as the SMRT+AFT batch.

2) LAP (line 408). Response: This is a typo error, should be “LSP” or Laser Shock Peening. The abbreviation of LSP was explained in an earlier sentence.

3) HABG (line 418). Response: This is also a typo, should be “HAGB” or high-angle-grain boundary. The abbreviation of HAGB was explained in an earlier sentence.

4) WPC (Fig. 6 and Fig. 15). Response: The WPC was the label for specimens treated with 55 µm shots. This is a fine particle shot peening process which is also known as the WPC treatment. The caption text for Figure 6 was revised to clarify the naming.

5) LCF (line 573). Response: The LCF stands for “low-cycle fatigue”, the text has now been corrected accordingly.

Comment 4: Figure numbers.

Figures on pages 10, 11, 12, and 13 have incorrect numbers.

Page 10, line 303: Fig 2 – the correct number is Fig. 8

Page 11, line 332: Fig. 3 – the correct number is Fig. 9

Page 12, line 376: Fig. 4 – the correct number is Fig. 10

Page 13, line 398: Fig. 5 – the correct number is Fig. 11

Response: We apologize for the figure numbering errors. It appears to be a formatting issue that occurred when the manuscript was uploaded, but we have verified again that they now show the correct figure numbers for the noted figures in the latest revision of the manuscript.

Comment 5: Figure captions.

If a figure shows microstructures or any other data gained by researching a specific material and a specific surface treatment, the figure caption should specify the materials designation, and the applied surface treatment. This was not the case at Fig. 5, Fig. 3 (page 11, the correct number would be Fig. 9), Fig. 4 (page 12, the correct number would be Fig. 10), Fig. 12, Fig. 13, and Fig. 14.

Response: Good feedback. The figure captions have been updated for Figure 5, Figure 9, Figure 10, Figure 12, Figure 13, and Figure 14. The missing material and surface treatment information has been added to the noted captions.

  1. The following figures are not mentioned in the text:

Fig. 1, Fig. 2, Fig. 3 and Fig. 4.

Response: The manuscript text has been revised to reference Figure 1, Figure 2, Figure 3, and Figure 4 at appropriate places in the text.

Reviewer 2 Report

The presented article is a good research of the influence of various surface strengthening technologies on fatigue life. The article lacks any of its own research on the issue. The article only describes and summarizes the results of other authors. 

It would be appropriate to add references to the figures to the text of Figures 1-4.
Image numbering has no sequence.

Author Response

Comment 1: The presented article is a good research of the influence of various surface strengthening technologies of fatigue life. The article lacks any of its own research on the issue. The article only describes and summarizes the results of other authors.

Response: Since this is a review article, the intent of the work is to provide a comparison of noted treatment methods based on results from other research and studies. Although no personal research was described, the content of this article should still be able to fulfill the goal of the work.

Comment 2: It would be appropriate to add reference to the figures to the text of Figures 1 - 4. Image numbering has no sequence.

Response: The manuscript text has been revised to reference Figure 1, Figure 2, Figure 3, and Figure 4 at appropriate places in the text. We apologize for the figure numbering errors. It appears to be a formatting issue that occurred when the manuscript was uploaded, but we have verified again that they now show the correct figure numbers in the latest revision of the manuscript.

Reviewer 3 Report

The paper concerns a comparison review study on fatigue resistance of metals subjected to different surface treatment methods. The paper is structured and clearly written focusing on the principal parameters, affecting the fatigue life, such as surface finish, microstructure and residual stress. The subject of the manuscript is within the scope of Metals and it can be considered for publication further to suggested following revisions and comments:

  1. The Figures used in the manuscript have to be original and un-published, unless prior license/permission to use is already granted.
  2. There are many abbreviations referred (e.g. SP, LSP, SMAT, UNSM, etc.). A list of explanation of abbreviations is recommended to be introduced (as a notation or list of symbols/abbreviations).
  3. As a general remark, please add a more detailed comment concerning the distinction of surface and internal fatigue crack initiation mechanism(s). See for instance Figure 5, describing also more clearly the type of open and solid symbols in the S-N curves.
  4. Comparing HCF vs. VHCF which is the typical boundary (Number of Cycles?) that stands as a limit between the two regimes?
  5. In Table 2, please explain what does the column “Specimens” represent.
  6. The graphs included in Figures 7b,c are not well understood. Is there any comparison involving the “initial” fatigue life (before any surface treatment) included? The percentages, in Figures 7b and c, have to be explained in higher detail.
  7. Please correct the numbering of Figures (from 8 to 11). After Figure 7 (from Fig. 8 to Fig. 11), the corresponding Figure numbers (referred in their captions) have been altered by mistake.
  8. In page 17 (Line 536), the reference could be probably incorrect. Maybe, Figure 6 has to be referred (instead of Figure 9). Please check and advise.
  9. Please explain in higher detail the diagrams included in Figures 15 and 16 (e.g. stress levels in the legend boxes).
  10. The inclusion of further recent References is suggested if they are available.
  11. The grammar/language is sufficient. However, a final proofreading is suggested to eliminate minor spelling errors. For instance, in page 16 (Lines 507-508) it is written: “As shown in this two research” – please correct this sentence (e.g. As shown in these two References [ , ]).

Author Response

Comment 1: The Figures used in the manuscript have to be original and un-published, unless prior license/permission to use is already granted.

Response: The licenses and permissions for the referenced Figures have already been obtained and provided to the MDPI.

Comment 2: There are many abbreviations referred (e.g. SP, LSP, SMAT, UNSM, etc.) A list of explanation of abbreviations is recommended to be introduced (as a notation or list of symbols/abbreviations).

Response: A list of abbreviations and explanations have been added to the manuscript just before the Introduction section.

Comment 3: As a general remark, please add a more detailed comment concerning the distinction of surface and internal fatigue crack initiation mechanism(s). See for instance Figure 5, describing also more clearly the type of open and solid symbols in the S-N curves.

Response:

1) The meanings of open solid symbols should be clarified. Their meanings have now been added to the Figure 5 caption.

2) It’s indeed that the surface-crack and interior-crack initiation mechanisms were not directly commented on or discussed in detail at places where the fatigue test results are first presented in Section 3, but they are added in other sections in the article. Because the focus of this article is to discuss the effects of surface treatments on fatigue properties of metals in HCF and VHCF regimes, the discussions and comments about surface-crack and interior-crack initiation mechanisms are also specifically separated into different sessions to support the aim of this article.

First of all, an introduction to the general surface-crack and interior-crack initiations is discussed in the second paragraph of the Introduction session. This paragraph gives readers some general key points: (1) the general differences between surface-crack and interior-crack initiation mechanisms; (2) surface-crack and interior-crack failures can lead to significant differences in fatigue lives; (3) surface treatments strengthen or weaken fatigue properties of metals in HCF and VHCF regimes by promoting or delaying surface-crack and interior-crack fractures. As per the recommendation, we have added sentences to clearly highlight the above key points. These key points serve as the background information to help readers better understand the rest of the article, but they are not the focus of this work, so we didn’t repeat them again later in the article.

In Section 3, fatigue test results showing the effects of different surface treatments on fatigue properties of metal in HCF and VHCF regimes were presented. Some simple comments about the occurrence of surface-crack or interior-crack failures were made, but they are not extensively discussed in this section. This is because there are many factors that can influence whether surface-crack or interior-crack failures will occur. The material differences are major factors. Differences in yield and tensile strengths of raw materials can influence the failure mode. Some materials, such as additively manufactured metals tend to have more internal defects and inclusions, which are more likely to have interior-crack failures. The fatigue test types also make difference in failure modes. The rotary bending fatigue test induces less fatigue stress in the interior of test specimens than in the surface of test specimens; therefore, interior-crack failures are generally less likely to occur than surface-crack failures during rotary-bending tests. These factors noted above, however, are not part of the effects caused by surface treatments, so they were not discussed in this article to minimize confusion.

Furthermore, the treatment-induced influences on surface-crack or interior-crack failure modes are generally caused by the combined effects of treatment-induced changes on surface finish, microstructure and microhardness, and residual stresses. Thus, detailed comments or comparisons about the failure modes focusing on the referenced studies can be rather difficult or confusing if presented in Section 3. It’s more clear and better organized to discuss how the treatment-induced changes in surface finish, microstructure and microhardness, and residual stresses affect surface-crack and interior-crack fractures differently in the following Section 4, 5, and 6.

For instance, the fatigue test results shown in Figure 5 were from the research by Dong et al. [1] based on SMRT-treated FSW 7075-T651 aluminum specimens. In paragraph 2 of Section 4, we mentioned that having a smooth surface finish can suppress surface-crack initiation based on the test results shown in Figure 5. To make additional comments, we now also added that interior-crack initiation and propagation were not directly affected by surface finish conditions. In Section 5, we commented that the work-hardened layer can suppress crack initiation in the surface region, and gradient microstructure and nanograin layer can also hinder surface crack initiation and slow crack growth. In this section, we noted that the SMRT-treated FSW 7075-T651 aluminum specimens showed both of these beneficial features after treatments. In Section 6, we noted that treatment-induced compressive residual stress prevents crack initiation in the compressive zone. This helped to improve the fatigue properties of SMRT-treated FSW 7075-T651 aluminum specimens in the research by Dong et al. [1] because surface-crack failures were observed before treatment. we did add another sentence here to clearly point out the cause-and-effect relationship.

Comment 4: Comparing HCF vs. VHCF which is the typical boundary (Number of Cycles?) that stands as a limit between the two regimes?

Response: Yes, HCF vs VHCF boundaries were not previously mentioned. The following sentences have been added to the manuscript Introduction section to clarify the difference between the two regimes: “The HCF regime typically refers to the region between 104 to 107 fatigue load cycles; whereas the VHCF regime typically refers to the region beyond 107 fatigue load cycles. Metal materials generally have higher fatigue stress limits in the HCF regime than in the VHCF regime.”

Comment 5: In Table 2, please explain what does the column “Specimens” represent.

Response: In Table 2, the column “Specimens” is supposed to represent the conditions of the specimens which also correspond to the symbol labels in Figure 7. The column title is now revised to “specimen condition”. We hope this will help to clarify the meaning of the column.

Comment 6: The graphs included in Figures 7b, c are not well understood. Is there any comparison involving the “initial” fatigue life (before any surface treatment) included? The percentages, in Figures 7b and c, have to be explained in higher detail.

Response: In the referenced article, the as-received material condition is the electropolished condition (labeled as “EP”). The author of referenced article only compared the 7075-T6 aluminum fatigue lives after SMAT treatment against the fatigue life of the material in electropolished surface conditions. We also noticed that the initial comments in Figure 7 were not sufficient and not as accurate to an extent. We have revised the comments about Figure 7 in Section 3.2 to the following:

According to the research from Gao et al. [15], SMAT had positive effects on fatigue lives of 7075-T6 aluminum specimens subjected to uniaxial fatigue load at the stress level of 220 MPa, but negative effects were observed at the stress level of 180 MPa (reference Table 2 and Figure 7). The as-received specimens were electropolished and they are referred to as the EP batch. Specimens from the Steel-2 and Steel-3 batches were subjected to different SMAT treatments, and specimens from the Steel-3MP batch had additional post-polishing than the Steel-3 batch specimens. At the fatigue stress level of 220 MPa, the specimens from the Steel-2 batch had an average 193% longer fatigue life than the EP batch, and the specimen from the Steel-3 batch had a 154% longer fatigue life than the EP batch (reference Figure 7 (b)). At the fatigue stress level of 180 MPa, the specimens from the Steel-2 batch had an average 68% shorter fatigue life than the EP batch, and the specimen from the Steel-3 batch had an average 95% shorter fatigue life than the EP batch (reference Figure 7 (c)).”

Comment 7: Please correct the numbering of Figures (from 8 to 11). After Figure 7 (from Fig. 8 to Fig. 11), the corresponding Figure numbers (referred in their captions) have been altered by mistake.

Response: We apologize for the figure numbering errors. It appears to be a formatting issue that occurred when the manuscript was uploaded, but we have verified again that they now show the correct figure numbers for the noted figures in the latest revision of the manuscript.

Comment 8: In page 17 (Line 536), the reference could be probably incorrect. Maybe, Figure 6 has to be referred (instead of Figure 9). Please check and advise.

Response: This is indeed an error. The error is now corrected in the last revision of the manuscript.

Comment 9: Please explain in higher detail the diagrams included in Figures 15 and 16 (e.g. stress levels in the legend boxes).

Response: We agree that Figures 15 and Figure 16 were not well explained. We have revised the text and added more explanations and clarifications such as the following:

“The cyclic yield strengths for raw 35CrMo and LZ50 steel specimens were 705 MPa and 313 MPa, respectively. Figure 15 shows the relaxation curves of compressive residual stress in microshot-peened 35CrMo steel specimens at 600 MPa and 650 MPa fatigue stress levels, and the relaxation curve of compressive residual stress in microshot-peened LZ50 steel specimens at 260 MPa fatigue stress levels. These fatigue stress levels were lower than the material cyclic yield strengths, so the compressive residual stresses quickly relaxed in the first few cycles and then became relatively stable. Figure 16 shows relaxation curves of compressive residual stress in microshot-peened LZ50 steel specimens at 380 MPa and 360 MPa fatigue stress levels. These fatigue stress levels were higher than the material yield strength, so the compressive residual stresses for LZ50 specimens continued to relax until approximately Nf = 104 cycles before stabilizing. It’s worth noting that the compressive residual stress was almost completely relaxed at the fatigue stress level of 380 MPa.”

Comment 10: The inclusion of further recent References is suggested if they are available.

Response:  Yes, added a comment about interior-crack failure with the white rough area (WRA) in the Introduction section based on the reference article from Xu et al. [2]. In Section 5.2, We also added a brief comment about gradient microstructure on workpieces treated with SP, SMAT, and LSP, and the referenced articles are from Skowron et al. [3], Zhou et al. [4], Yang et al. [5], and Pour-Ali et al. [6].

Comment 11: The grammar/language is sufficient. However, a final proofreading is suggested to eliminate minor spelling errors. For instance, in page 16 (Lines 507-508) it is written: “As shown in this two research” – please correct this sentence (as shown in these two references [,]).

Response: We did struggle a little with the particular sentence noted in the comment because “research” is understood to be a mass word. We have revised the text from “research” to “references” per the recommendation. Additionally, we will do our best to proofread the manuscript again to check for similar issues.

Above are our responses and explanations to the suggestions and comments.

Thank you again for the support and understanding.

Reference:

  1. Dong, P.; Liu, Z.; Zhai, X.; Yan, Z.; Wang, W.; Liaw, P.K. Incredible Improvement in Fatigue Resistance of Friction Stir Welded 7075-T651 Aluminum Alloy via Surface Mechanical Rolling Treatment. Int. J. Fatigue2019, 124, 15–25, doi:10.1016/j.ijfatigue.2019.02.023.
  2. Xu, L.; Wang, Q.; Zhou, M. Micro-Crack Initiation and Propagation in a High Strength Aluminum Alloy during Very High Cycle Fatigue. Mater. Sci. Eng. A2018, 715, 404–413, doi:10.1016/j.msea.2018.01.008.
  3. Skowron, K.; Wróbel, M.; MosiaÅ‚ek, M.; Joncour, L.L.; Dryzek, E. Gradient Microstructure Induced by Surface Mechanical Attrition Treatment in Grade 2 Titanium Studied Using Positron Annihilation Spectroscopy and Complementary Methods. Materials2021, 14, 6347, doi:10.3390/ma14216347.
  4. Zhou, J.; Sun, Z.; Kanouté, P.; Retraint, D. Effect of Surface Mechanical Attrition Treatment on Low Cycle Fatigue Properties of an Austenitic Stainless Steel. Int. J. Fatigue2017, 103, 309–317, doi:10.1016/j.ijfatigue.2017.06.011.
  5. Yang, Y.; Zhang, H.; Qiao, H. Microstructure Characteristics and Formation Mechanism of TC17 Titanium Alloy Induced by Laser Shock Processing. J. Alloys Compd.2017, 722, 509–516, doi:10.1016/j.jallcom.2017.06.127.
  6. Pour-Ali, S.; Kiani-Rashid, A.-R.; Babakhani, A.; Virtanen, S.; Allieta, M. Correlation between the Surface Coverage of Severe Shot Peening and Surface Microstructural Evolutions in AISI 321: A TEM, FE-SEM and GI-XRD Study. Surf. Coat. Technol.2018, 334, 461–470, doi:10.1016/j.surfcoat.2017.11.062.

Round 2

Reviewer 2 Report

The publication for revision meets the requirements for a review article. I agree to the publication only as a review article.

Reviewer 3 Report

The authors replied to the review comments and amended properly their manuscript which could be considered as acceptable publication.

Please during proofreading correct the X-axis marks in Figure 7b; Steel instead of Steal.